# Variational Inference for Gaussian Process Models with Linear Complexity

**Ching-An Cheng**
Institute for Robotics and Intelligent Machines
Georgia Institute of Technology
Atlanta, GA 30332
cacheng@gatech.edu

**Byron Boots**
Institute for Robotics and Intelligent Machines
Georgia Institute of Technology
Atlanta, GA 30332
bboots@cc.gatech.edu

## Abstract

Large-scale Gaussian process inference has long faced practical challenges due to time and space complexity that is superlinear in dataset size. While sparse variational Gaussian process models are capable of learning from large-scale data, standard strategies for sparsifying the model can prevent the approximation of complex functions. In this work, we propose a novel variational Gaussian process model that decouples the representation of mean and covariance functions in reproducing kernel Hilbert space. We show that this new parametrization generalizes previous models. Furthermore, it yields a variational inference problem that can be solved by stochastic gradient ascent with time and space complexity that is only linear in the number of mean function parameters, regardless of the choice of kernels, likelihoods, and inducing points. This strategy makes the adoption of large-scale expressive Gaussian process models possible. We run several experiments on regression tasks and show that this decoupled approach greatly outperforms previous sparse variational Gaussian process inference procedures.

## 1 Introduction

Gaussian process (GP) inference is a popular nonparametric framework for reasoning about functions under uncertainty. However, the expressiveness of GPs comes at a price: solving (approximate) inference for a GP with $N$ data instances has time and space complexities in $\Theta(N^3)$ and $\Theta(N^2)$, respectively. Therefore, GPs have traditionally been viewed as a tool for problems with small- or medium-sized datasets

Recently, the concept of inducing points has been used to scale GPs to larger datasets. The idea is to summarize a full GP model with statistics on a sparse set of $M \ll N$ fictitious observations [18, 24]. By representing a GP with these inducing points, the time and the space complexities are reduced to $O(NM^2 + M^3)$ and $O(NM + M^2)$, respectively. To further process datasets that are too large to fit into memory, stochastic approximations have been proposed for regression [10] and classification [11]. These methods have similar complexity bounds, but with $N$ replaced by the size of a mini-batch $N_m$.

Despite the success of sparse models, the scalability issues of GP inference are far from resolved. The major obstruction is that the cubic complexity in $M$ in the aforementioned upper-bound is also a lower-bound, which results from the inversion of an $M$-by-$M$ covariance matrix defined on the inducing points. As a consequence, these models can only afford to use a small set of $M$ basis functions, limiting the expressiveness of GPs for prediction.

In this work, we show that superlinear complexity is not completely necessary. Inspired by the reproducing kernel Hilbert space (RKHS) representation of GPs [2], we propose a generalized variational GP model, called DGPs (Decoupled Gaussian Processes), which *decouples* the bases

|  | **a, B** | $\alpha,\beta$ | $\theta$ | $\alpha = \beta$ | $N \neq M$ | Time | Space |
|---|---|---|---|---|---|---|---|
| SVDGP | SGA | SGA | SGA | FALSE | TRUE | $O(DNM_\alpha + NM_\beta^2 + M_\beta^3)$ | $O(NM_\alpha + M_\beta^2)$ |
| SVI | SNGA | SGA | SGA | TRUE | TRUE | $O(DNM + NM^2 + M^3)$ | $O(NM + M^2)$ |
| iVSGPR | SMA | SMA | SGA | TRUE | TRUE | $O(DNM + NM^2 + M^3)$ | $O(NM + M^2)$ |
| VSGPR | CG | CG | CG | TRUE | TRUE | $O(DNM + NM^2 + M^3)$ | $O(NM + M^2)$ |
| GPR | CG | CG | CG | TRUE | FALSE | $O(DN^2 + N^3)$ | $O(N^2)$ |

Table 1: Comparison between SVDGP and variational GPR algorithms: SVI [10], iVSGPR [2], VSGPR [24], and GPR [19], where $N$ is the number of observations/the size of a mini-batch, $M$, $M_\alpha$, $M_\beta$ are the number of basis functions, and $D$ is the input dimension. Here it is assumed $M_\alpha \geq M_\beta$ [1].

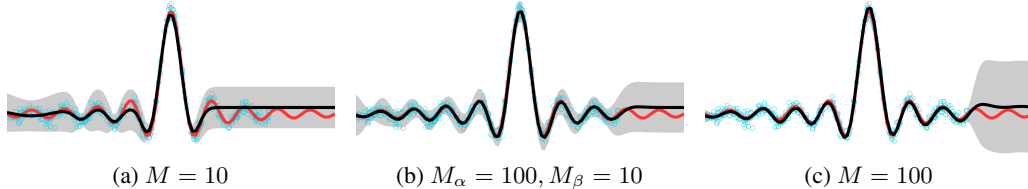

(a) $M = 10$  (b) $M_\alpha = 100, M_\beta = 10$  (c) $M = 100$

Figure 1: Comparison between models with shared and decoupled basis. (a)(c) denote the models with shared basis of size $M$. (b) denotes the model of decoupled basis with size $(M_\alpha, M_\beta)$. In each figure, the red line denotes the ground truth; the blue circles denote the observations; the black line and the gray area denote the mean and variance in prediction, respectively.

for the mean and the covariance functions. Specifically, let $M_\alpha$ and $M_\beta$ be the numbers of basis functions used to model the mean and the covariance functions, respectively. Assume $M_\alpha \geq M_\beta$. We show, when DGPs are used as a variational posterior [24], the associated variational inference problem can be solved by stochastic gradient ascent with space complexity $O(N_m M_\alpha + M_\beta^2)$ and time complexity $O(DN_m M_\alpha + N_m M_\beta^2 + M_\beta^3)$, where $D$ is the input dimension. We name this algorithm SVDGP. As a result, we can choose $M_\alpha \gg M_\beta$, which allows us to keep the time and space complexity similar to previous methods (by choosing $M_\beta = M$) while greatly increasing accuracy. To the best of our knowledge, this is the first variational GP algorithm that admits linear complexity in $M_\alpha$, without any assumption on the choice of kernel and likelihood.

While we design SVDGP for general likelihoods, in this paper we study its effectiveness in Gaussian process regression (GPR) tasks. We consider this is without loss of generality, as most of the sparse variational GPR algorithms in the literature can be modified to handle general likelihoods by introducing additional approximations (e.g. in Hensman et al. [11] and Sheth et al. [22]). Our experimental results show that SVDGP significantly outperforms the existing techniques, achieving higher variational lower bounds and lower prediction errors when evaluated on held-out test sets.

## 1.1 Related Work

Our framework is based on the variational inference problem proposed by Titsias [24], which treats the inducing points as variational parameters to allow direct approximation of the true posterior. This is in contrast to Seeger et al. [21], Snelson and Ghahramani [23], Quiñonero-Candela and Rasmussen [18], and Lázaro-Gredilla et al. [15], which all use inducing points as hyper-parameters of a degenerate prior. While both approaches have the same time and space complexity, the latter additionally introduces a large set of unregularized hyper-parameters and, therefore, is more likely to suffer from over-fitting [1].

In Table 1, we compare SVDGP with recent GPR algorithms in terms of the assumptions made and the time and space complexity. Each algorithm can be viewed as a special way to solve the maximization of the variational lower bound (5), presented in Section 3.2. Our algorithm SVDGP generalizes the previous approaches to allow the basis functions for the mean and the covariance to be decoupled, so an approximate solution can be found by stochastic gradient ascent in linear complexity.

To illustrate the idea, we consider a toy GPR example in Figure 1. The dataset contains 500 noisy observations of a *sinc* function. Given the same training data, we conduct experiments with three different GP models. Figure 1 (a)(c) show the results of the traditional coupled basis, which can be solved by any of the variational algorithms listed in Table 1, and Figure 1 (b) shows the result using the decoupled approach SVDGP. The sizes of basis and observations are selected to emulate a large dataset scenario. We can observe SVDGP achieves a nice trade-off between prediction performance and complexity: it achieves almost the same accuracy in prediction as the full-scale model in Figure 1(c) and preserves the overall shape of the predictive variance.

In addition to the sparse algorithms above, some recent attempts aim to revive the non-parametric property of GPs by structured covariance functions. For example, Wilson and Nickisch [27] proposes to space the inducing points on a multidimensional lattice, so the time and space complexities of using a product kernel becomes $O(N + DM^{1+1/D})$ and $O(N + DM^{1+2/D})$, respectively. However, because $M = c^D$, where $c$ is the number of grid points per dimension, the overall complexity is exponential in $D$ and infeasible for high-dimensional data. Another interesting approach by Hensman et al. [12] combines variational inference [24] and a sparse spectral approximation [15]. By equally spacing inducing points on the spectrum, they show the covariance matrix on the inducing points have diagonal plus low-rank structure. With MCMC, the algorithm can achieve complexity $O(DNM)$. However, the proposed structure in [12] does not help to reduce the complexity when an approximate Gaussian posterior is favored or when the kernel hyper-parameters need to be updated.

Other kernel methods with linear complexity have been proposed using functional gradient descent [14, 5]. However, because these methods use a model strictly the same size as the entire dataset, they fail to estimate the predictive covariance, which requires $\Omega(N^2)$ space complexity. Moreover, they cannot learn hyper-parameters online. The latter drawback also applies to greedy algorithms based on rank-one updates, e.g. the algorithm of Csató and Opper [4].

In contrast to these previous methods, our algorithm applies to *all* choices of inducing points, likelihoods, and kernels, and we allow both variational parameters and hyper-parameters to adapt online as more data are encountered.

## 2 Preliminaries

In this section, we briefly review the inference for GPs and the variational framework proposed by Titsias [24]. For now, we will focus on GPR for simplicity of exposition. We will discuss the case of general likelihoods in the next section when we introduce our framework, DGPs.

### 2.1 Inference for GPs

Let $f : \mathcal{X} \to \mathbb{R}$ be a latent function defined on a compact domain $\mathcal{X} \subset \mathbb{R}^D$. Here we assume *a priori* that $f$ is distributed according to a Gaussian process $\mathcal{GP}(m, k)$. That is, $\forall x, x' \in \mathcal{X}$, $\mathbb{E}[f(x)] = m(x)$ and $\mathbb{C}[f(x), f(x')] = k(x, x')$. In short, we write $f \sim \mathcal{GP}(m, k)$.

A GP probabilistic model is composed of a likelihood $p(y|f(x))$ and a GP prior $\mathcal{GP}(m, k)$; in GPR, the likelihood is assumed to be Gaussian i.e. $p(y|f(x)) = \mathcal{N}(y|f(x), \sigma^2)$ with variance $\sigma^2$. Usually, the likelihood and the GP prior are parameterized by some hyper-parameters, which we summarize as $\theta$. This includes, for example, the variance $\sigma^2$ and the parameters implicitly involved in defining $k(x, x')$. For notational convenience, and without loss of generality, we assume $m(x) = 0$ in the prior distribution and omit explicitly writing the dependence of distributions on $\theta$.

Assume we are given a dataset $\mathcal{D} = \{(x_n, y_n)\}_{n=1}^N$, in which $x_n \in \mathcal{X}$ and $y_n \sim p(y|f(x_n))$. Let[2] $X = \{x_n\}_{n=1}^N$ and $\mathbf{y} = (y_n)_{n=1}^N$. Inference for GPs involves solving for the posterior $p_{\theta*}(f(x)|\mathbf{y})$ for any new input $x \in \mathcal{X}$, where $\theta^* = \arg\max_\theta \log p_\theta(\mathbf{y})$. For example in GPR, because the likelihood is Gaussian, the predictive posterior is also Gaussian with mean and covariance

$$m_{|\mathbf{y}}(x) = \mathbf{k}_{x,X}(\mathbf{K}_X + \sigma^2 \mathbf{I})^{-1}\mathbf{y}, \qquad k_{|\mathbf{y}}(x, x') = k_{x,x'} - \mathbf{k}_{x,X}(\mathbf{K}_X + \sigma^2 \mathbf{I})^{-1}\mathbf{k}_{X,x'}, \quad (1)$$

and the hyper-parameter $\theta^*$ can be found by nonlinear conjugate gradient ascent [19]

$$\max_\theta \log p_\theta(\mathbf{y}) = \max_\theta \log \mathcal{N}(\mathbf{y}|0, \mathbf{K}_X + \sigma^2 \mathbf{I}), \tag{2}$$

where $k_{\cdot,\cdot}$, $\mathbf{k}_{\cdot,\cdot}$ and $\mathbf{K}_{\cdot,\cdot}$ denote the covariances between the sets in the subscript.[3] One can show that these two functions, $m_{|\mathbf{y}}(x)$ and $k_{|\mathbf{y}}(x, x')$, define a valid GP. Therefore, given observations $\mathbf{y}$, we say $f \sim \mathcal{GP}(m_{|\mathbf{y}}, k_{|\mathbf{y}})$.

Although theoretically GPs are non-parametric and can model any function as $N \to \infty$, in practice this is difficult. As the inference has time complexity $\Omega(N^3)$ and space complexity $\Omega(N^2)$, applying vanilla GPs to large datasets is infeasible.

## 2.2 Variational Inference with Sparse GPs

To scale GPs to large datasets, Titsias [24] introduced a scheme to compactly approximate the true posterior with a sparse GP, $\mathcal{GP}(\hat{m}_{|\mathbf{y}}, \hat{k}_{|\mathbf{y}})$, defined by the statistics on $M \ll N$ function values: $\{L_m f(\tilde{x}_m)\}_{m=1}^M$, where $L_m$ is a bounded linear operator[4] and $\tilde{x}_m \in \mathcal{X}$. $L_m f(\cdot)$ is called an *inducing function* and $\tilde{x}_m$ an *inducing point*. Common choices of $L_m$ include the identity map (as used originally by Titsias [24]) and integrals to achieve better approximation or to consider multi-domain information [26, 7, 3]. Intuitively, we can think of $\{L_m f(\tilde{x}_m)\}_{m=1}^M$ as a set of potentially indirect observations that capture salient information about the unknown function $f$.

Titsias [24] solves for $\mathcal{GP}(\hat{m}_{|\mathbf{y}}, \hat{k}_{|\mathbf{y}})$ by variational inference. Let $\tilde{X} = \{\tilde{x}_m\}_{m=1}^M$ and let $\mathbf{f}_X \in \mathbb{R}^N$ and $\mathbf{f}_{\tilde{X}} \in \mathbb{R}^M$ be the (inducing) function values defined on $X$ and $\tilde{X}$, respectively. Let $p(\mathbf{f}_{\tilde{X}})$ be the prior given by $\mathcal{GP}(m, k)$ and define $q(\mathbf{f}_{\tilde{X}}) = \mathcal{N}(\mathbf{f}_{\tilde{X}} | \tilde{\mathbf{m}}, \tilde{\mathbf{S}})$ to be its variational posterior, where $\tilde{\mathbf{m}} \in \mathbb{R}^M$ and $\tilde{\mathbf{S}} \in \mathbb{R}^{M \times M}$ are the mean and the covariance of the approximate posterior of $\mathbf{f}_{\tilde{X}}$. Titsias [24] proposes to use $q(\mathbf{f}_X, \mathbf{f}_{\tilde{X}}) = p(\mathbf{f}_X | \mathbf{f}_{\tilde{X}}) q(\mathbf{f}_{\tilde{X}})$ as the variational posterior to approximate $p(\mathbf{f}_X, \mathbf{f}_{\tilde{X}} | \mathbf{y})$ and to solve for $q(\mathbf{f}_{\tilde{X}})$ together with the hyper-parameter $\theta$ through

$$\max_{\theta, \tilde{X}, \tilde{\mathbf{m}}, \tilde{\mathbf{S}}} \mathcal{L}_\theta(\tilde{X}, \tilde{\mathbf{m}}, \tilde{\mathbf{S}}) = \max_{\theta, \tilde{X}, \tilde{\mathbf{m}}, \tilde{\mathbf{S}}} \int q(\mathbf{f}_X, \mathbf{f}_{\tilde{X}}) \log \frac{p(\mathbf{y}|\mathbf{f}_X) p(\mathbf{f}_X|\mathbf{f}_{\tilde{X}}) p(\mathbf{f}_{\tilde{X}})}{q(\mathbf{f}_X, \mathbf{f}_{\tilde{X}})} \mathrm{d}\mathbf{f}_X \mathrm{d}\mathbf{f}_{\tilde{X}}, \quad (3)$$

where $\mathcal{L}_\theta$ is a variational lower bound of $\log p_\theta(\mathbf{y})$, $p(\mathbf{f}_X | \mathbf{f}_{\tilde{X}}) = \mathcal{N}(\mathbf{f}_X | \mathbf{K}_{X,\tilde{X}} \mathbf{K}_{\tilde{X}}^{-1} \mathbf{f}_{\tilde{X}}, \mathbf{K}_X - \hat{\mathbf{K}}_X)$ is the conditional probability given in $\mathcal{GP}(m, k)$, and $\hat{\mathbf{K}}_X = \mathbf{K}_{X,\tilde{X}} \mathbf{K}_{\tilde{X}}^{-1} \mathbf{K}_{\tilde{X},X}$.

At first glance, the specific choice of variational posterior $q(\mathbf{f}_X, \mathbf{f}_{\tilde{X}})$ seems heuristic. However, although parameterized finitely, it resembles a full-fledged GP $\mathcal{GP}(\hat{m}_{|\mathbf{y}}, \hat{k}_{|\mathbf{y}})$:

$$\hat{m}_{|\mathbf{y}}(x) = \mathbf{k}_{x,\tilde{X}} \mathbf{K}_{\tilde{X}}^{-1} \tilde{\mathbf{m}}, \qquad \hat{k}_{|\mathbf{y}}(x, x') = k_{x,x'} + \mathbf{k}_{x,\tilde{X}} \mathbf{K}_{\tilde{X}}^{-1} \left( \tilde{\mathbf{S}} - \mathbf{K}_{\tilde{X}} \right) \mathbf{K}_{\tilde{X}}^{-1} \mathbf{k}_{\tilde{X},x'}. \quad (4)$$

This result is further studied in Matthews et al. [16] and Cheng and Boots [2], where it is shown that (3) is indeed minimizing a proper KL-divergence between Gaussian processes/measures.

By comparing (2) and (3), one can show that the time and the space complexities now reduce to $O(DNM + M^2N + M^3)$ and $O(M^2 + MN)$, respectively, due to the low-rank structure of $\hat{\mathbf{K}}_{\tilde{X}}$ [24]. To further reduce complexity, stochastic optimization, such as stochastic natural ascent [10] or stochastic mirror descent [2] can be applied. In this case, $N$ in the above asymptotic bounds would be replaced by the size of a mini-batch $N_m$. The above results can be modified to consider general likelihoods as in [22, 11].

## 3 Variational Inference with Decoupled Gaussian Processes

Despite the success of sparse GPs, the scalability issues of GPs persist. Although parameterizing a GP with inducing points/functions enables learning from large datasets, it also restricts the expressiveness of the model. As the time and the space complexities still scale in $\Omega(M^3)$ and $\Omega(M^2)$, we cannot learn or use a complex model with large $M$.

In this work, we show that these two complexity bounds, which have long accompanied GP models, are not strictly necessary, but are due to the tangled representation canonically used in the GP

literature. To elucidate this, we adopt the dual representation of Cheng and Boots [2], which treats GPs as linear operators in RKHS. But, unlike Cheng and Boots [2], we show how to decouple the basis representation of mean and covariance functions of a GP and derive a new variational problem, which can be viewed as a generalization of (3). We show that this problem—with arbitrary likelihoods and kernels—can be solved by stochastic gradient ascent with linear complexity in $M_\alpha$, the number of parameters used to specify the mean function for prediction.

In the following, we first review the results in [2]. We next introduce the decoupled representation, DGPs, and its variational inference problem. Finally, we present SVDGP and discuss the case with general likelihoods.

## 3.1 Gaussian Processes as Gaussian Measures

Let an RKHS $\mathcal{H}$ be a Hilbert space of functions with the reproducing property: $\forall x \in \mathcal{X}$, $\exists \phi_x \in \mathcal{H}$ such that $\forall f \in \mathcal{H}$, $f(x) = \phi_x^T f$.[5] A Gaussian process $\mathcal{GP}(m, k)$ is equivalent to a Gaussian measure $\nu$ on Banach space $\mathcal{B}$ which possesses an RKHS $\mathcal{H}$ [2]:[6] there is a mean functional $\mu \in \mathcal{H}$ and a bounded positive semi-definite linear operator $\Sigma : \mathcal{H} \to \mathcal{H}$, such that for any $x, x' \in \mathcal{X}$, $\exists \phi_x, \phi_{x'} \in \mathcal{H}$, we can write $m(x) = \phi_x^T \mu$ and $k(x, x') = \phi_x^T \Sigma \phi_{x'}$. The triple $(\mathcal{B}, \nu, \mathcal{H})$ is known as an abstract Wiener space [9, 6], in which $\mathcal{H}$ is also called the Cameron-Martin space. Here the restriction that $\mu, \Sigma$ are RKHS objects is necessary, so the variational inference problem in the next section can be well-defined.

We call this the *dual* representation of a GP in RKHS $\mathcal{H}$ (the mean function $m$ and the covariance function $k$ are realized as linear operators $\mu$ and $\Sigma$ defined in $\mathcal{H}$). With abuse of notation, we write $\mathcal{N}(f|\mu, \Sigma)$ in short. This notation does not mean a GP has a Gaussian distribution in $\mathcal{H}$, nor does it imply that the sample paths from $\mathcal{GP}(m, k)$ are necessarily in $\mathcal{H}$. Precisely, $\mathcal{B}$ contains the sample paths of $\mathcal{GP}(m, k)$ and $\mathcal{H}$ is dense in $\mathcal{B}$. In most applications of GP models, $\mathcal{B}$ is the Banach space of continuous function $C(\mathcal{X}; \mathcal{Y})$ and $\mathcal{H}$ is the span of the covariance function. As a special case, if $\mathcal{H}$ is finite-dimensional, $\mathcal{B}$ and $\mathcal{H}$ coincide and $\nu$ becomes equivalent to a Gaussian distribution in a Euclidean space.

In relation to our previous notation in Section 2.1: suppose $k(x, x') = \phi_x^T \phi_{x'}$ and $\phi_x : \mathcal{X} \to \mathcal{H}$ is a feature map to some Hilbert space $\mathcal{H}$. Then we have assumed *a priori* that $\mathcal{GP}(m, k) = \mathcal{N}(f|0, I)$ is a normal Gaussian measure; that is $\mathcal{GP}(m, k)$ samples functions $f$ in the form $f(x) = \sum_{l=1}^{\dim \mathcal{H}} \phi_l(x)^T \epsilon_l$, where $\epsilon_l \sim \mathcal{N}(0, 1)$ are independent. Note if $\dim \mathcal{H} = \infty$, with probability one $f$ is not in $\mathcal{H}$, but fortunately $\mathcal{H}$ is large enough for us to approximate the sampled functions. In particular, it can be shown that the posterior $\mathcal{GP}(m_{|\mathbf{y}}, k_{|\mathbf{y}})$ in GPR has a dual RKHS representation in the same RKHS as the prior GP [2].

## 3.2 Variational Inference in Gaussian Measures

Cheng and Boots [2] proposes a dual formulation of (3) in terms of Gaussian measures[7]:

$$\max_{q(f),\theta} \mathcal{L}_\theta(q(f)) = \max_{q(f),\theta} \int q(f) \log \frac{p_\theta(y|f)p(f)}{q(f)} \mathrm{d}f = \max_{q(f),\theta} \mathbb{E}_q[\log p_\theta(y|f)] - \mathrm{KL}[q||p], \quad (5)$$

where $q(f) = \mathcal{N}(f|\tilde{\mu}, \tilde{\Sigma})$ is a variational Gaussian measure and $p(f) = \mathcal{N}(f|0, I)$ is a normal prior. Its connection to the inducing points/functions in (3) can be summarized as follows [2, 3]: Define a linear operator $\Psi_{\tilde{X}} : \mathbb{R}^M \to \mathcal{H}$ as $\mathbf{a} \mapsto \sum_{m=1}^{M} a_m \psi_{\tilde{x}_m}$, where $\psi_{\tilde{x}_m} \in \mathcal{H}$ is defined such that $\psi_{\tilde{x}_m}^T \mu = \mathbb{E}[L_m f(\tilde{x}_m)]$. Then (3) and (5) are equivalent, if $q(f)$ has a *subspace parametrization*,

$$\tilde{\mu} = \Psi_{\tilde{X}} \mathbf{a}, \quad \tilde{\Sigma} = I + \Psi_{\tilde{X}} \mathbf{A} \Psi_{\tilde{X}}^T, \quad (6)$$

with $\mathbf{a} \in \mathbb{R}^M$ and $\mathbf{A} \in \mathbb{R}^{M \times M}$ satisfying $\tilde{\mathbf{m}} = \mathbf{K}_{\tilde{X}} \mathbf{a}$, and $\tilde{\mathbf{S}} = \mathbf{K}_{\tilde{X}} + \mathbf{K}_{\tilde{X}} \mathbf{A} \mathbf{K}_{\tilde{X}}$. In other words, the variational inference algorithms in the literature are all using a variational Gaussian measure in which $\tilde{\mu}$ and $\tilde{\Sigma}$ are parametrized by the same basis $\{\psi_{\tilde{x}_m} | \tilde{x}_m \in \tilde{X}\}_{i=1}^{M}$.

Compared with (3), the formulation in (5) is neater: it follows the definition of the very basic variational inference problem. This is not surprising, since GPs can be viewed as Bayesian linear models in an infinite-dimensional space. Moreover, in (5) all hyper-parameters are isolated in the likelihood $p_\theta(y|f)$, because the prior is fixed as a normal Gaussian measure.

### 3.3 Disentangling the GP Representation with DGPs

While Cheng and Boots [2] treat (5) as an equivalent form of (3), here we show that it is a generalization. By further inspecting (5), it is apparent that sharing the basis $\Psi_{\tilde{X}}$ between $\tilde{\mu}$ and $\tilde{\Sigma}$ in (6) is not strictly necessary, since (5) seeks to optimize two linear operators, $\tilde{\mu}$ and $\tilde{\Sigma}$. With this in mind, we propose a new parametrization that *decouples* the bases for $\tilde{\mu}$ and $\tilde{\Sigma}$:

$$\tilde{\mu} = \Psi_\alpha \mathbf{a}, \quad \tilde{\Sigma} = (I + \Psi_\beta \mathbf{B} \Psi_\beta^T)^{-1} \tag{7}$$

where $\Psi_\alpha : \mathbb{R}^{M_\alpha} \to \mathcal{H}$ and $\Psi_\beta : \mathbb{R}^{M_\beta} \to \mathcal{H}$ denote linear operators defined similarly to $\Psi_{\tilde{X}}$ and $\mathbf{B} \succeq 0 \in \mathbb{R}^{M_\beta \times M_\beta}$. Compared with (6), here we parametrize $\tilde{\Sigma}$ through its inversion with $\mathbf{B}$ so the condition that $\tilde{\Sigma} \succeq 0$ can be easily realized as $\mathbf{B} \succeq 0$. This form agrees with the posterior covariance in GPR [2] and will give a posterior that is strictly less uncertain than the prior. Note the choice of decoupled parametrization is not unique. In particular, the bases can be partially shared, or $(\mathbf{a}, \mathbf{B})$ can be further parametrized (e.g. $\mathbf{B}$ can be parametrized using the canonical form in (4)) to improve the numerical convergence rate. Please refer to Appendix A for a discussion.[8]

The decoupled subspace parametrization (7) corresponds to a DGP, $\mathcal{GP}(\hat{m}_{|\mathbf{y}}^\alpha, \hat{k}_{|\mathbf{y}}^\beta)$, with mean and covariance functions as [9]

$$\hat{m}_{|\mathbf{y}}^\alpha(x) = \mathbf{k}_{x,\alpha} \boldsymbol{a}, \qquad \hat{k}_{|\mathbf{y}}^\beta(x, x') = k_{x,x'} - \mathbf{k}_{x,\beta} \left( \mathbf{B}^{-1} + \mathbf{K}_\beta \right)^{-1} \mathbf{k}_{\beta,x'}. \tag{8}$$

While the structure of (8) looks similar to (4), directly replacing the basis $\tilde{X}$ in (4) with $\alpha$ and $\beta$ is not trivial. Because the equations in (4) are derived from the traditional viewpoint of GPs as statistics on function values, the original optimization problem (3) is not defined if $\alpha \neq \beta$ and therefore, it is not clear how to learn a decoupled representation traditionally. Conversely, by using the dual RKHS representation, the objective function to learn (8) follows naturally from (5), as we will show next.

### 3.4 SVDGP: Algorithm and Analysis

Substituting the decoupled subspace parametrization (7) into the variational inference problem in (5) results in a numerical optimization problem: $\max_{q(f),\theta} \mathbb{E}_q[\log p_\theta(y|f)] - \text{KL}[q||p]$ with

$$\text{KL}[q||p] = \frac{1}{2}\mathbf{a}^T \mathbf{K}_\alpha \mathbf{a} + \frac{1}{2}\log|\mathbf{I} + \mathbf{K}_\beta \mathbf{B}| + \frac{-1}{2}\text{tr}\left(\mathbf{K}_\beta(\mathbf{B}^{-1} + \mathbf{K}_\beta)^{-1}\right) \tag{9}$$

$$\mathbb{E}_q[\log p_\theta(y|f)] = \sum_{n=1}^N \mathbb{E}_{q(f(x_n))}[\log p_\theta(y_n|f(x_n))] \tag{10}$$

where each expectation is over a scalar Gaussian $q(f(x_n))$ given by (8) as functions of $(\mathbf{a}, \alpha)$ and $(\mathbf{B}, \beta)$. Our objective function contains [11] as a special case, which assumes $\alpha = \beta = \tilde{X}$. In addition, we note that Hensman et al. [11] indirectly parametrize the posterior by $\tilde{\mathbf{m}}$ and $\tilde{\mathbf{S}} = \mathbf{L}\mathbf{L}^T$, whereas we parametrize directly by (6) with $\mathbf{a}$ for scalability and $\mathbf{B} = \tilde{\mathbf{L}}\mathbf{L}^T$ for better stability (which always reduces the uncertainty in the posterior compared with the prior).

We notice that $(\mathbf{a}, \alpha)$ and $(\mathbf{B}, \beta)$ are completely decoupled in (9) and *potentially* combined again in (10). In particular, if $p_\theta(y_n|f(x_n))$ is Gaussian as in GPR, we have an additional decoupling, i.e. $\mathcal{L}_\theta(\mathbf{a}, \mathbf{B}, \alpha, \beta) = \mathcal{F}_\theta(\mathbf{a}, \alpha) + \mathcal{G}_\theta(\mathbf{B}, \beta)$ for some $\mathcal{F}_\theta(\mathbf{a}, \alpha)$ and $\mathcal{G}_\theta(\mathbf{B}, \beta)$. Intuitively, the optimization

**Algorithm 1** Online Learning with DGPs

---

**Parameters:** $M_\alpha$, $M_\beta$, $N_m$, $N_\Delta$
**Input:** $\mathcal{M}(\mathbf{a}, \mathbf{B}, \alpha, \beta, \theta)$, $\mathcal{D}$
 1: $\theta_0 \leftarrow$ initializeHyperparameters( sampleMinibatch($\mathcal{D}, N_m$) )
 2: **for** $t = 1 \ldots T$ **do**
 3:    $D_t \leftarrow$ sampleMinibatch($\mathcal{D}, N_m$)
 4:    $\mathcal{M}$.addBasis($D_t$, $N_\Delta$, $M_\alpha$, $M_\beta$)
 5:    $\mathcal{M}$.updateModel($D_t$, t)
 6: **end for**

---

over $(\mathbf{a}, \alpha)$ aims to minimize the fitting-error, and the optimization over $(\mathbf{B}, \beta)$ aims to memorize the samples encountered so far; the mean and the covariance functions only interact indirectly through the optimization of the hyper-parameter $\theta$.

One salient feature of SVDGP is that it tends to overestimate, rather than underestimate, the variance, when we select $M_\beta \leq M_\alpha$. This is inherited from the non-degeneracy property of the variational framework [24] and can be seen in the toy example in Figure 1. In the extreme case when $M_\beta = 0$, we can see the covariance in (8) becomes the same as the prior; moreover, the objective function of SVDGP becomes similar to kernel methods (exactly the same as kernel ridge regression, when the likelihood is Gaussian). The additional inclusion of expected log-likelihoods here allows SVDGP to learn the hyper-parameters in a unified framework, as its objective function can be viewed as minimizing a generalization upper-bound in PAC-Bayes learning [8].

SVDGP solves the above optimization problem by stochastic gradient ascent. Here we purposefully ignore specific details of $p_\theta(y|f)$ to emphasize that SVDGP can be applied to general likelihoods as it only requires unbiased first-order information, which e.g. can be found in [22]. In addition to having a more adaptive representation, the main benefit of SVDGP is that the computation of an unbiased gradient requires only linear complexity in $M_\alpha$, as shown below (see Appendix B for details).

**KL-Divergence** Assume $|\alpha| = O(DM_\alpha)$ and $|\beta| = O(DM_\beta)$. By (9), One can show $\nabla_\mathbf{a} \mathrm{KL}[q||p] = \mathbf{K}_\alpha \mathbf{a}$ and $\nabla_\mathbf{B} \mathrm{KL}[q||p] = \frac{1}{2}(\mathbf{I} + \mathbf{K}_\beta \mathbf{B})^{-1} \mathbf{K}_\beta \mathbf{B} \mathbf{K}_\beta (\mathbf{I} + \mathbf{B} \mathbf{K}_\beta)^{-1}$. Therefore, the time complexity to compute $\nabla_\mathbf{a} \mathrm{KL}[q||p]$ can be reduced to $O(N_m M_\alpha)$ if we sample over the columns of $\mathbf{K}_\alpha$ with a mini-batch of size $N_m$. By contrast, the time complexity to compute $\nabla_\mathbf{B} \mathrm{KL}[q||p]$ will always be $\Theta(M_\beta^3)$ and cannot be further reduced, regardless of the parametrization of $\mathbf{B}$.[10] The gradient with respect to $\alpha$ and $\beta$ can be derived similarly and have time complexity $O(DN_m M_\alpha)$ and $O(DM_\beta^2 + M_\beta^3)$, respectively.

**Expected Log-Likelihood** Let $\hat{\mathbf{m}}(\mathbf{a}, \alpha) \in \mathbb{R}^N$ and $\hat{\mathbf{s}}(\mathbf{B}, \beta) \in \mathbb{R}^N$ be the vectors of the mean and covariance of scalar Gaussian $q(f(x_n))$ for $n \in \{1, \ldots, N\}$. As (10) is a sum over $N$ terms, by sampling with a mini-batch of size $N_m$, an unbiased gradient of (10) with respect to $(\theta, \hat{\mathbf{m}}, \hat{\mathbf{s}})$ can be computed in $O(N_m)$. To compute the full gradient with respect to $(\mathbf{a}, \mathbf{B}, \alpha, \beta)$, we compute the derivative of $\hat{\mathbf{m}}$ and $\hat{\mathbf{s}}$ with respect to $(\mathbf{a}, \mathbf{B}, \alpha, \beta)$ and then apply chain rule. These steps take $O(DN_m M_\alpha)$ and $O(DN_m M_\beta + N_m M_\beta^2 + M_\beta^3)$ for $(\mathbf{a}, \alpha)$ and $(\mathbf{B}, \beta)$, respectively.

The above analysis shows that the curse of dimensionality in GPs originates in the covariance function. For space complexity, the decoupled parametrization (7) requires memory in $O(N_m M_\alpha + M_\beta^2)$; for time complexity, an unbiased gradient with respect to $(\mathbf{a}, \alpha)$ can be computed in $O(DN_m M_\alpha)$, but that with respect to $(\mathbf{B}, \beta)$ has time complexity $\Omega(DN_m M_\beta + N_m M_\beta^2 + M_\beta^3)$. This motivates choosing $M_\beta = O(M)$ and $M_\alpha$ in $O(M_\beta^2)$ or $O(M_\beta^3)$, which maintains the same complexity as previous variational techniques but greatly improves the prediction performance.

## 4 Experimental Results

We compare our new algorithm, SVDGP, with the state-of-the-art incremental algorithms for sparse variational GPR, SVI [10] and iVSGPR [2], as well as the classical GPR and the batch algorithm VS-GPR [24]. As discussed in Section 1.1, these methods can be viewed as different ways to optimize (5). Therefore, in addition to the normalized mean square error (nMSE) [19] in prediction, we report

| KUKA$_1$ - Variational Lower Bound ($10^5$) | | | | |
| --- | --- | --- | --- | --- |
| | SVDGP | SVI | iVSGPR | VSGPR | GPR |
| mean | **1.262** | 0.391 | 0.649 | 0.472 | -5.335 |
| std | **0.195** | 0.076 | 0.201 | 0.265 | 7.777 |

| KUKA$_1$ - Prediction Error (nMSE) | | | | |
| --- | --- | --- | --- | --- |
| | SVDGP | SVI | iVSGPR | VSGPR | GPR |
| mean | **0.037** | 0.169 | 0.128 | 0.139 | 0.231 |
| std | **0.013** | 0.025 | 0.033 | 0.026 | 0.045 |

| MUJOCO$_1$ - Variational Lower Bound ($10^5$) | | | | |
| --- | --- | --- | --- | --- |
| | SVDGP | SVI | iVSGPR | VSGPR | GPR |
| mean | **6.007** | 2.178 | 4.543 | 2.822 | -10312.727 |
| std | **0.673** | 0.692 | 0.898 | 0.871 | 22679.778 |

| MUJOCO$_1$ - Prediction Error (nMSE) | | | | |
| --- | --- | --- | --- | --- |
| | SVDGP | SVI | iVSGPR | VSGPR | GPR |
| mean | **0.072** | 0.163 | 0.099 | 0.118 | 0.213 |
| std | **0.013** | 0.053 | 0.026 | 0.016 | 0.061 |

Table 2: Experimental results of KUKA$_1$ and MUJOCO$_1$ after 2,000 iterations.

the performance in the variational lower bound (VLB) (5), which also captures the quality of the predictive variance and hyper-parameter learning.[11] These two metrics are evaluated on held-out test sets in all of our experimental domains.

Algorithm 1 summarizes the online learning procedure used by all stochastic algorithms,[12] where each learner has to optimize all the parameters on-the-fly using *i.i.d.* data. The hyper-parameters are first initialized heuristically by median trick using the first mini-batch. We incrementally build up the variational posterior by including $N_\Delta \leq N_m$ observations in each mini-batch as the *initialization* of new variational basis functions. Then all the hyper-parameters and the variational parameters are updated online. These steps are repeated for $T$ iterations.

For all the algorithms, we assume the prior covariance is defined by the SE-ARD kernel [19] and we use the generalized SE-ARD kernel [2] as the inducing functions in the variational posterior (see Appendix C for details). We note that all algorithms in comparison use the same kernel and optimize both the variational parameters (including inducing points) and the hyperparameters.

In particular, we implement SGA by ADAM [13] (with default parameters $\beta_1 = 0.9$ and $\beta_2 = 0.999$). The step-size for each stochastic algorithms is scheduled according to $\gamma_t = \gamma_0(1 + 0.1\sqrt{t})^{-1}$, where $\gamma_0 \in \{10^{-1}, 10^{-2}, 10^{-3}\}$ is selected manually for each algorithm to maximize the improvement in objective function after the first 100 iterations. We test each stochastic algorithm for $T = 2000$ iterations with mini-batches of size $N_m = 1024$ and the increment size $N_\Delta = 128$. Finally, the model sizes used in the experiments are listed as follows: $M_\alpha = 128^2$ and $M_\beta = 128$ for SVDGP; $M = 1024$ for SVI; $M = 256$ for iVSGPR; $M = 1024$, $N = 4096$ for VSGPR; $N = 1024$ for GP. These settings share similar order of time complexity in our current Matlab implementation.

## 4.1 Datasets

**Inverse Dynamics of KUKA Robotic Arm** This dataset records the inverse dynamics of a KUKA arm performing rhythmic motions at various speeds [17]. The original dataset consists of two parts: KUKA$_1$ and KUKA$_2$, each of which have 17,560 offline data and 180,360 online data with 28 attributes and 7 outputs. In the experiment, we mix the online and the offline data and then split 90% as training data (178,128 instances) and 10% testing data (19,792 instances) to satisfy the *i.i.d.* assumption.

**Walking MuJoCo** MuJoCo (Multi-Joint dynamics with Contact) is a physics engine for research in robotics, graphics, and animation, created by [25]. In this experiment, we gather 1,000 walking trajectories by running TRPO [20]. In each time frame, the MuJoCo transition dynamics have a 23-dimensional input and a 17-dimensional output. We consider two regression problems to predict 9 of the 17 outputs from the input[13]: MUJOCO$_1$ which maps the input of the current frame (23 dimensions) to the output, and MUJOCO$_2$ which maps the inputs of the current and the previous frames (46 dimensions) to the output. In each problem, we randomly select 90% of the data as training data (842,745 instances) and 10% as test data (93,608 instances).

## 4.2 Results

We summarize part of the experimental results in Table 2 in terms of nMSE in prediction and VLB. While each output is treated independently during learning, Table 2 present the mean and the standard

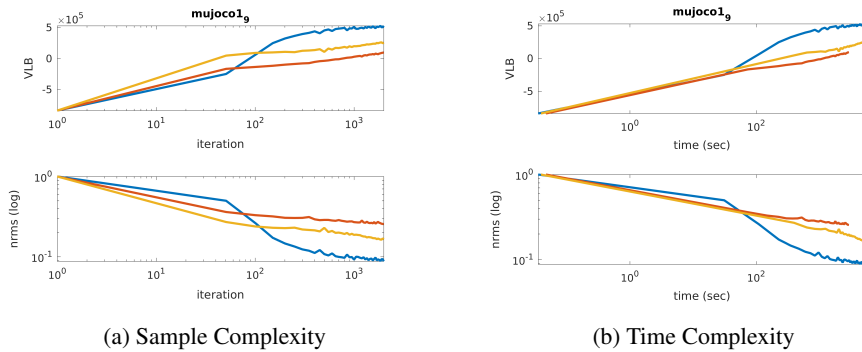

(a) Sample Complexity           (b) Time Complexity

Figure 2: An example of online learning results (the 9th output of MUJOCO$_1$ dataset). The blue, red, and yellow lines denote SVDGP, SVI, and iVSGPR, respectively.

deviation over all the outputs as the selected metrics are normalized. For the complete experimental results, please refer to Appendix D.

We observe that SVDGP consistently outperforms the other approaches with much higher VLBs and much lower prediction errors; SVDGP also has smaller standard deviation. These results validate our initial hypothesis that adopting a large set of basis functions for the mean can help when modeling complicated functions. iVSGPR has the next best result after SVDGP, despite using a basis size of 256, much smaller than that of 1,024 in SVI, VSGPR, and GPR. Similar to SVDGP, iVSGPR also generalizes better than the batch algorithms VSGPR and GPR, which only have access to a smaller set of training data and are more prone to over-fitting. By contrast, the performance of SVI is surprisingly worse than VSGPR. We conjecture this might be due to the fact that the hyper-parameters and the inducing points/functions are only crudely initialized in online learning. We additionally find that the stability of SVI is more sensitive to the choice of step size than other methods. This might explain why in [10, 2] batch data was used to initialize the hyper-parameters and the learning rate to update the hyper-parameters was selected to be much smaller than that for stochastic natural gradient ascent.

To further investigate the properties of different stochastic approximations, we show the change of VLB and the prediction error over iterations and time in Figure 2. Overall, whereas iVSGPR and SVI share similar convergence rate, the behavior of SVDGP is different. We see that iVSGPR converges the fastest, both in time and sample complexity. Afterwards, SVDGP starts to descend faster and surpass the other two methods. From Figure 2, we can also observe that although SVI has similar convergence to iVSGPR, it slows down earlier and therefore achieves a worse result. These phenomenon are observed in multiple experiments.

## 5   Conclusion

We propose a novel, fully-differentiable framework, Decoupled Gaussian Processes DGPs, for large-scale GP problems. By decoupling the representation, we derive a variational inference problem that can be solved with stochastic gradients with *linear* time and space complexity. Compared with existing algorithms, SVDGP can adopt a much larger set of basis functions to predict more accurately. Empirically, SVDGP significantly outperforms state-of-the-arts variational sparse GPR algorithms in multiple regression tasks. These encouraging experimental results motivate further application of SVDGP to end-to-end learning with neural networks in large-scale, complex real world problems.

### Acknowledgments

This work was supported in part by NSF NRI award 1637758. The authors additionally thank the reviewers and Hugh Salimbeni for productive discussion which improved the quality of the paper.

## Footnotes

[1]The first three columns show the algorithms to update the parameters: SGA/SNGA/SMA denotes stochastic gradient/natural gradient/mirror ascent, and CG denotes batch nonlinear conjugate gradient ascent. The 4th and the 5th columns indicate whether the bases for mean and covariance are strictly shared, and whether a variational posterior can be used. The last two columns list the time and space complexity.

[2] In notation, we use boldface to distinguish finite-dimensional vectors (lower-case) and matrices (upper-case) that are used in computation from scalar and abstract mathematical objects.

[3]If the two sets are the same, only one is listed.

[4]Here we use the notation $L_m f$ loosely for the compactness of writing. Rigorously, $L_m$ is a bounded linear operator acting on $m$ and $k$, not necessarily on all sample paths $f$.

[5]To simplify the notation, we write $\phi_x^T f$ for $\langle f, \phi_x \rangle_{\mathcal{H}}$, and $f^T L g$ for $\langle f, L g \rangle_{\mathcal{H}}$, where $f, g \in \mathcal{H}$ and $L : \mathcal{H} \to \mathcal{H}$, even if $\mathcal{H}$ is infinite-dimensional.

[6]Such $\mathcal{H}$ w.l.o.g. can be identified as the natural RKHS of the covariance function of a zero-mean prior GP.

[7] We assume $q(f)$ is absolutely continuous wrt $p(f)$, which is true as $p(f)$ is non-degenerate. The integral denotes the expectation of $\log p_\theta(y|f) + \log \frac{p(f)}{q(f)}$ over $q(f)$, and $\frac{q(f)}{p(f)}$ denotes the Radon-Nikodym derivative.

[8]Appendix A is partially based on a discussion with Hugh Salimbeni at the NIPS conference. Here we adopt the fully decoupled, directly parametrized form in (7) to demonstrate the idea. We leave the full comparison of different decoupled parametrizations in future work.

[9]In practice, we can parametrize $\mathbf{B} = \mathbf{L}\mathbf{L}^T$ with Cholesky factor $\mathbf{L} \in \mathbb{R}^{M_\beta \times M_\beta}$ so the problem is unconstrained. The required terms in (8) and later in (9) can be stably computed as $\left(\mathbf{B}^{-1} + \mathbf{K}_\beta\right)^{-1} = \mathbf{L}\mathbf{H}^{-1}\mathbf{L}^T$ and $\log|\mathbf{I} + \mathbf{K}_\beta \mathbf{B}| = \log|\mathbf{H}|$, where $\mathbf{H} = \mathbf{I} + \mathbf{L}^T \mathbf{K}_\beta \mathbf{L}$.

[10]Due to $\mathbf{K}_\beta$, the complexity would remain as $O(M_\beta^3)$ even if $\mathbf{B}$ is constrained to be diagonal.

[11]The exact marginal likelihood is computationally infeasible to evaluate for our large model.

[12]The algorithms differs only in whether the bases are shared and how the model is updated (see Table 1).

[13]Because of the structure of MuJoCo dynamics, the rest 8 outputs can be trivially known from the input.

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
