[Supplementary Material · appendix.pdf]

# Appendix

## A    Subspace Parametrization and Notes to Practitioners

As mentioned in Section 3.3, the choice of subspace parametrization is non-unique and not limited to

$$\tilde{\mu} = \Psi_\alpha \mathbf{a}, \quad \tilde{\Sigma} = (I + \Psi_\beta \mathbf{B} \Psi_\beta^T)^{-1}.$$

While we adopted the completely decoupled, directly parametrized representation in the experiments for its simplicity and sufficiency to validate our idea, this choice may not possess the best numerical properties. Here we point out other potential, practical parameterizations. A complete study of these choices is outside of the scope of this paper.

### A.1    Numerical Convergence Issues

To understand the effect of the parametrization on the convergence rate, intuition can be gained by inspecting the objective function $\max_{q(f),\theta} \mathbb{E}_q[\log p_\theta(y|f)] - \text{KL}[q||p]$, where

$$\text{KL}[q||p] = \frac{1}{2}\mathbf{a}^T \mathbf{K}_\alpha \mathbf{a} + \frac{1}{2}\log|\mathbf{I} + \mathbf{K}_\beta \mathbf{B}| + \frac{-1}{2}\text{tr}\left(\mathbf{K}_\beta(\mathbf{B}^{-1} + \mathbf{K}_\beta)^{-1}\right)$$

$$\mathbb{E}_q[\log p_\theta(y|f)] = \sum_{n=1}^{N} \mathbb{E}_{q(f(x_n))}[\log p_\theta(y_n|f(x_n))]$$

When the likelihood function is Gaussian, this leads to a problem which is quadratic in $\mathbf{a}$. Therefore, the numerical convergence rate can be slow, for example, when $\mathbf{K}_\alpha$ is ill-conditioned (which can happen especially when $M_\alpha$ is large and $\Psi_\alpha$ is not flexible and correlated). Empirically, we observed a slow-down of convergence rate when the standard SE-ARD kernel was used as the variational basis, compared with the generalized SE-ARD kernel used in the experiments.

The use of a preconditioner can help the convergence of first-order methods in practice. This can be achieved by further parameterizing $(\mathbf{a}, \mathbf{B})$. For example, a Jacobi preconditioner can be optionally used in implementation by parameterizing $\mathbf{a}$ and $\mathbf{L}$ (i.e. $\mathbf{B} = \mathbf{L}\mathbf{L}^T$) through $\mathbf{a}_0$ and $\mathbf{L}_0$ as

$$\mathbf{a} = \text{diag}(\mathbf{K}_\alpha)^{-1}\mathbf{a}_0, \qquad \mathbf{L} = \text{diag}(\mathbf{K}_\beta)^{-1}\mathbf{L}_0$$

where $\text{diag}$ denotes the diagonal part. While the Jacobi preconditioner is only $\rho^{-2}\mathbf{I}$ for SE-ARD kernels for some scaling constant $\rho \in \mathbb{R}$, we observed it still helps the convergence rate as the hyperparameter $\rho$ is also being updated online.

Comparing the Jacobi preconditioner and (4), we can see that the canonical parametrization in (4) can be viewed as a full preconditioner, which can enjoy a better numerical convergence rate but at the cost of higher computational complexity. For the mean function, this type of full preconditioner, with $\mathbf{K}_\alpha^{-1}$, is too expensive to compute when $M_\alpha$ is large; nonetheless, using an approximation, such as the Nyström approximation or (incomplete) Cholesky decomposition, can potentially improve convergence without increasing the computational complexity. For the covariance function, because $M_\beta$ is small, a full preconditioner can be used, such as setting $\mathbf{L} = \mathbf{K}_\beta^{-1}\mathbf{L}_0$ or parameterizing $\mathbf{B}$ with the canonical choice (4) (see Appendix A.3 for a further discussion).

### A.2    Non-convexity and Initialization

Since the optimization problem is non-convex, the performance of the model also hinges on how the variational basis functions are initialized. We note that, when using the completely decoupled parametrization as in the experiments, we advocate to initialize the new basis for the mean and the covariance to be the same samples (e.g. from the current mini-batch) and then updating them separately online. This would encourage $\tilde{\mu}$ and $\tilde{\Sigma}$ to capture the information in a close subspace.

Another idea to improve stability is to partially share the basis functions. For example, the first $M_\beta$ basis functions of the total $M_\alpha$ basis functions for the mean can be constrained to be the same as the basis functions for the covariance. Due to the redundancy in the mean parametrization, sharing part of the parameters can make the problem more well-conditioned and easier to optimize.

### A.3 Hybrid Subspace Parametrization

An example[14] that combines the ideas from the above two sections is a hybrid subspace parametrization of the variational Gaussian measure:

$$\tilde{\mu} = \Psi_{\tilde{X}} \mathbf{K}_{\tilde{X}}^{-1} \tilde{\mathbf{m}} + \Psi_{\alpha_r} \mathbf{a}_r \qquad \tilde{\Sigma} = I + \Psi_{\tilde{X}} \mathbf{K}_{\tilde{X}}^{-1} \left( \tilde{\mathbf{S}} - \mathbf{K}_{\tilde{X}} \right) \mathbf{K}_{\tilde{X}}^{-1} \Psi_{\tilde{X}}^T \tag{11}$$

where $\tilde{\mathbf{m}} \in \mathbb{R}^M$ and $\tilde{\mathbf{S}} \succeq 0 \in \mathbb{R}^{M \times M}$ are equivalent to the posterior statistics on $\tilde{X}$ used in the conventional shared representation, $\alpha_r$ denotes the additional inducing points that model the residual error, and $\mathbf{a}_r$ is the corresponding coefficient. This representation is equivalent to setting $\beta = \tilde{X}$ and $\mathbf{B}^{-1} = -(\mathbf{K}_{\tilde{X}} + \mathbf{K}_{\tilde{X}}(\tilde{\mathbf{S}} - \mathbf{K}_{\tilde{X}})^{-1}\mathbf{K}_{\tilde{X}})$. The hybrid subspace parametrization gives a predictive model in the form

$$\hat{m}_{|\mathbf{y}}^H(x) = \mathbf{k}_{x,\tilde{X}} \mathbf{K}_{\tilde{X}}^{-1} \tilde{\mathbf{m}} + \mathbf{k}_{x,\alpha_r} \mathbf{a}_r \tag{12}$$

$$\hat{k}_{|\mathbf{y}}^H(x, x') = k_{x,x'} + \mathbf{k}_{x,\tilde{X}} \mathbf{K}_{\tilde{X}}^{-1} \left( \tilde{\mathbf{S}} - \mathbf{K}_{\tilde{X}} \right) \mathbf{K}_{\tilde{X}}^{-1} \mathbf{k}_{\tilde{X},x'} \tag{13}$$

Compared with (4), the only difference is the residual term $\mathbf{k}_{x,\alpha_r} \mathbf{a}_r$, which helps modeling more complex functions.

Therefore, this new, hybrid formulation shows a more direct connection to the conventional parametrization used in the GP literature (e.g. [11]). This can also been seen in its associated objective function. Substitute the hybrid parametrization into the terms in the KL divergence between Gaussian measures and we have the following relations:

$$\frac{1}{2}\mathbf{a}^T \mathbf{K}_\alpha \mathbf{a} = \frac{1}{2}\tilde{\mathbf{m}}^T \mathbf{K}_{\tilde{X}}^{-1} \tilde{\mathbf{m}} + \tilde{\mathbf{m}}^T \mathbf{K}_{\tilde{X}}^{-1} \mathbf{K}_{\tilde{X},\alpha_r} \mathbf{a}_r + \frac{1}{2}\mathbf{a}_r^T \mathbf{K}_{\alpha_r} \mathbf{a}_r$$

$$\frac{-1}{2}\mathrm{tr}\left( \mathbf{K}_\beta (\mathbf{B}^{-1} + \mathbf{K}_\beta)^{-1} \right) = \frac{1}{2}\mathrm{tr}\left( \tilde{\mathbf{S}} \mathbf{K}_{\tilde{X}}^{-1} \right) - \frac{|\tilde{X}|}{2}$$

$$\frac{1}{2}\log |\mathbf{I} + \mathbf{K}_\beta \mathbf{B}| = -\log |\mathbf{K}_{\tilde{X}}^{-1} \tilde{\mathbf{S}}|.$$

Therefore, the KL divergence term for the hybrid parametrization can be written as

$$\mathrm{KL}[q||p] = \frac{1}{2}\tilde{\mathbf{m}}^T \mathbf{K}_{\tilde{X}}^{-1} \tilde{\mathbf{m}} + \tilde{\mathbf{m}}^T \mathbf{K}_{\tilde{X}}^{-1} \mathbf{K}_{\tilde{X},\alpha_r} \mathbf{a}_r + \frac{1}{2}\mathbf{a}_r^T \mathbf{K}_{\alpha_r} \mathbf{a}_r$$
$$- \log |\mathbf{K}_{\tilde{X}}^{-1} \tilde{\mathbf{S}}| + \frac{1}{2}\mathrm{tr}\left( \tilde{\mathbf{S}} \mathbf{K}_{\tilde{X}}^{-1} \right) - \frac{|\tilde{X}|}{2} \tag{14}$$

This KL divergence terms is exactly as the one used by Hensman et al. [11], when $\tilde{\mathbf{a}}_r = 0$. That is, the hybrid parametrization is a strict generalization of the canonical parametrization. Note in here $\tilde{\mathbf{S}}$ is initialized as $\tilde{\mathbf{S}} = \tilde{\mathbf{K}}_{\tilde{X}} = \mathbf{L}\mathbf{L}^T$ and then its Cholesky factor $\mathbf{L}$ is optimized afterwards.

From (12), (13), and (14), we can see the linear complexity of the decoupled model is preserved. The stochastic gradient can be computed in linear time by performing sampling of the residual inducing points $\alpha_r$.

## B   Variational Inference with Decoupled Gaussian Processes

Here we provide the details of the variational inference problem used to learn DGPs:

$$\max_{q(f),\theta} \mathcal{L}_\theta(q(f) = \max_{q(f),\theta} \int q(f) \log \frac{p_\theta(y|f)p(f)}{q(f)} \mathrm{d}f = \max_{q(f),\theta} \mathbb{E}_q[\log p_\theta(y|f)] - \mathrm{KL}[q||p], \tag{15}$$

### B.1 KL Divergence

#### B.1.1 Evaluation

First, we show how to evaluate the KL-divergence. We do so by extending the KL-divergence between two finite-dimensional subspace-parametrized Gaussian measures to infinite dimensional space and show that it is well-defined.

Recall for two $d$-dimensional Gaussian distributions $q(f) = \mathcal{N}(f|\mu, \Sigma)$ and $p = \mathcal{N}(f|\bar{\mu}, \bar{\Sigma})$, the KL-divergence is given as

**Proposition 1.**

$$\mathrm{KL}[q||p] := \int \log \frac{q(f)}{p(f)} \mathrm{d}\mu_q(f) = \int q(f) \log \frac{q(f)}{p(f)} \mathrm{d}f$$

$$= \frac{1}{2}\left(\mathrm{tr}\left(\bar{\Sigma}^{-1}\Sigma\right) + (\mu - \bar{\mu})^T \bar{\Sigma}^{-1}(\mu - \bar{\mu}) + \ln \frac{|\bar{\Sigma}|}{|\Sigma|} - d\right)$$

Now consider $q$ and $p$ are subspace parametrized as

$$p(f) = \mathcal{N}(f|\bar{\mu}, \bar{\Sigma}) = \mathcal{N}(f|\Psi_{\bar{\alpha}}\bar{\mathbf{a}}, (I + \Psi_{\bar{\beta}}\bar{\mathbf{B}}\Psi_{\bar{\beta}}^T)^{-1})$$
$$q(f) = \mathcal{N}(f|\mu, \Sigma) = \mathcal{N}(f|\Psi_{\alpha}\mathbf{a}, (I + \Psi_{\beta}\mathbf{B}\Psi_{\beta}^T)^{-1}). \tag{16}$$

By Proposition 1, we derive the representation of KL-divergence which is applicable even when $d$ is infinite. Recall in the infinite dimensional case, $\mu$, $\Sigma$, $\bar{\mu}$, and $\bar{\Sigma}$ are objects in the RKHS $\mathcal{H}$ (Cameron-Martin space).

**Theorem 1.** *Assume $q$ and $p$ are two subspace parametrized Gaussian measures given as* (16). *Regardless of the dimension of $\mathcal{H}$, the following holds*

$$\mathrm{KL}[q||p] = \frac{-1}{2}\mathrm{tr}\left(\left(\mathbf{K}_{\beta} + \mathbf{K}_{\beta,\bar{\beta}}\bar{\mathbf{B}}\mathbf{K}_{\bar{\beta},\beta}\right)\left(\mathbf{B}^{-1} + \mathbf{K}_{\beta}\right)^{-1}\right) + \frac{1}{2}\log|\mathbf{I} + \mathbf{K}_{\beta}\mathbf{B}|$$

$$+ \frac{1}{2}\mathbf{a}^T\left(\mathbf{K}_{\alpha} + \mathbf{K}_{\alpha,\bar{\beta}}\bar{\mathbf{B}}\mathbf{K}_{\bar{\beta},\alpha}\right)\mathbf{a} - \mathbf{a}^T\left(\mathbf{K}_{\alpha,\bar{\alpha}} + \mathbf{K}_{\alpha,\bar{\beta}}\bar{\mathbf{B}}\mathbf{K}_{\bar{\beta},\bar{\alpha}}\right)\bar{\mathbf{a}} + C \tag{17}$$

*where*

$$C = \frac{1}{2}\left(\mathrm{tr}\left(\mathbf{K}_{\bar{\beta}}\bar{\mathbf{B}}\right) - \log|\mathbf{I} + \mathbf{K}_{\bar{\beta}}\bar{\mathbf{B}}| + \bar{\mathbf{a}}^T\left(\mathbf{K}_{\bar{\alpha}} + \mathbf{K}_{\bar{\alpha},\bar{\beta}}\bar{\mathbf{B}}\mathbf{K}_{\bar{\beta},\bar{\alpha}}\right)\bar{\mathbf{a}}\right)$$

*In particular, if $p$ is normal (i.e. $p(f) = \mathcal{N}(f|0, I)$), then*

$$\mathrm{KL}[q||p] = \frac{1}{2}\mathbf{a}^T\mathbf{K}_{\alpha}\mathbf{a} + \frac{1}{2}\log|\mathbf{I} + \mathbf{K}_{\beta}\mathbf{B}| + \frac{-1}{2}\mathrm{tr}\left(\mathbf{K}_{\beta}(\mathbf{B}^{-1} + \mathbf{K}_{\beta})^{-1}\right)$$

*Proof.*
To prove, we derive each term in (17) as follows.

First, we derive $\mathrm{tr}\left(\bar{\Sigma}^{-1}\Sigma\right) - d$. Define $\mathbf{R} = (\mathbf{B}^{-1} + \mathbf{K}_{\beta})^{-1}$. Then we can write

$$\Sigma = (I + \Psi_{\beta}\mathbf{B}\Psi_{\beta}^T)^{-1} = I - \Psi_{\beta}(\mathbf{B}^{-1} + \Psi_{\beta}^T\Psi_{\beta})^{-1}\Psi_{\beta}^T = I - \Psi_{\beta}\mathbf{R}\Psi_{\beta}^T. \tag{18}$$

Using (18), we can derive

$$\bar{\Sigma}^{-1}\Sigma = (I + \Psi_{\bar{\beta}}\bar{\mathbf{B}}\Psi_{\bar{\beta}}^T)\left(I - \Psi_{\beta}\mathbf{R}\Psi_{\beta}^T\right) = I + \Psi_{\bar{\beta}}\bar{\mathbf{B}}\Psi_{\bar{\beta}}^T - \Psi_{\beta}\mathbf{R}\Psi_{\beta}^T - \Psi_{\bar{\beta}}\bar{\mathbf{B}}\mathbf{K}_{\bar{\beta},\beta}\mathbf{R}\Psi_{\beta}^T$$

and therefore

$$\mathrm{tr}\left(\bar{\Sigma}^{-1}\Sigma\right) - d = \mathrm{tr}\left(I\right) - d + \mathrm{tr}\left(\mathbf{K}_{\bar{\beta}}\bar{\mathbf{B}}\right) - \mathrm{tr}\left(\mathbf{R}\left(\mathbf{K}_{\beta} + \mathbf{K}_{\beta,\bar{\beta}}\bar{\mathbf{B}}\mathbf{K}_{\bar{\beta},\beta}\right)\right)$$
$$= \mathrm{tr}\left(\mathbf{K}_{\bar{\beta}}\bar{\mathbf{B}}\right) - \mathrm{tr}\left(\mathbf{R}\left(\mathbf{K}_{\beta} + \mathbf{K}_{\beta,\bar{\beta}}\bar{\mathbf{B}}\mathbf{K}_{\bar{\beta},\beta}\right)\right)$$

Note this term does not depend on the ambient dimension.

Second, we derive $\log(|\bar{\Sigma}|/|\Sigma|)$: Since

$$\log|\Sigma^{-1}| = \log|\mathbf{B}^{-1} + \mathbf{K}_{\beta}||\mathbf{B}| = \log|\mathbf{I} + \mathbf{K}_{\beta}\mathbf{B}|.$$

it holds that

$$\log \frac{|\bar{\Sigma}|}{|\Sigma|} = \log |\mathbf{I} + \mathbf{K}_\beta \mathbf{B}| - \log |\mathbf{I} + \mathbf{K}_{\bar{\beta}} \bar{\mathbf{B}}|.$$

Finally, we derive the quadratic term:

$$
\begin{aligned}
&(\mu - \bar{\mu})^T \bar{\Sigma}^{-1} (\mu - \bar{\mu}) \\
&= \mu^T \bar{\Sigma}^{-1} \mu - 2\bar{\mu}^T \bar{\Sigma}^{-1} \mu + \bar{\mu} \bar{\Sigma}^{-1} \bar{\mu} \\
&= \mathbf{a}^T \Psi_\alpha^T \left( I + \Psi_{\bar{\beta}} \bar{\mathbf{B}} \Psi_{\bar{\beta}}^T \right) \Psi_\alpha \mathbf{a} - 2\bar{\mathbf{a}}^T \Psi_{\bar{\alpha}}^T \left( I + \Psi_{\bar{\beta}} \bar{\mathbf{B}} \Psi_{\bar{\beta}}^T \right) \Psi_\alpha \mathbf{a} + \bar{\mathbf{a}}^T \Psi_{\bar{\alpha}}^T \left( I + \Psi_{\bar{\beta}} \bar{\mathbf{B}} \Psi_{\bar{\beta}}^T \right) \Psi_{\bar{\alpha}} \bar{5}\mathbf{a} \\
&= \mathbf{a}^T \left( \mathbf{K}_\alpha + \mathbf{K}_{\alpha,\bar{\beta}} \bar{\mathbf{B}} \mathbf{K}_{\bar{\beta},\alpha} \right) \mathbf{a} - 2\bar{\mathbf{a}}^T \left( \mathbf{K}_{\bar{\alpha},\alpha} + \mathbf{K}_{\bar{\alpha},\bar{\beta}} \bar{\mathbf{B}} \mathbf{K}_{\bar{\beta},\alpha} \right) \mathbf{a} + \bar{\mathbf{a}}^T \left( \mathbf{K}_{\bar{\alpha}} + \mathbf{K}_{\bar{\alpha},\bar{\beta}} \bar{\mathbf{B}} \mathbf{K}_{\bar{\beta},\bar{\alpha}} \right) \bar{\mathbf{a}}
\end{aligned}
$$

$\square$

**Remarks** The above expression is well defined even when $\mathbf{B} \succeq 0$, because $(\mathbf{B}^{-1} + \mathbf{K}_\beta)^{-1} = \mathbf{B}(\mathbf{I} + \mathbf{K}_\beta \mathbf{B})^{-1}$. Particularly, we can parametrize $\mathbf{B} = \mathbf{L}\mathbf{L}^T$ with Cholesky factor $\mathbf{L} \in \mathbb{R}^{M_\beta \times M_\beta}$ in practice so the problem is unconstrained. The required terms can be stably computed: $\left(\mathbf{B}^{-1} + \mathbf{K}_\beta\right)^{-1} = \mathbf{L}\mathbf{H}^{-1}\mathbf{L}^T$ and $\log |\mathbf{I} + \mathbf{K}_\beta \mathbf{B}| = \log |\mathbf{H}|$, where $\mathbf{H} = \mathbf{I} + \mathbf{L}^T \mathbf{K}_\beta \mathbf{L}$.

### B.1.2 Gradients

Here we derive the equations of the gradient of the variational inference problem of SVDGP. The purpose here is to show the complexity of calculating the gradients. These equations are useful in implementing SVDGP using basic linear algebra routines, while computational-graph libraries based on automatic differentiation are also applicable and easier to apply.

To derive the gradients, we first introduce some short-hand

$$
\begin{aligned}
\mathbf{G}_\alpha &= \mathbf{K}_\alpha + \mathbf{K}_{\alpha,\bar{\beta}} \bar{\mathbf{B}} \mathbf{K}_{\bar{\beta},\alpha} \\
\mathbf{G}_{\alpha,\bar{\alpha}} &= \mathbf{K}_{\alpha,\bar{\alpha}} + \mathbf{K}_{\alpha,\bar{\beta}} \bar{\mathbf{B}} \mathbf{K}_{\bar{\beta},\bar{\alpha}} \\
\mathbf{G}_\beta &= \mathbf{K}_\beta + \mathbf{K}_{\beta,\bar{\beta}} \bar{\mathbf{B}} \mathbf{K}_{\bar{\beta},\beta}
\end{aligned}
$$

and write $\mathrm{KL}[q||p]$ as

$$\mathrm{KL}[q||p] = \frac{-1}{2} \mathrm{tr}\left( \mathbf{G}_\beta (\mathbf{B}^{-1} + \mathbf{K}_\beta)^{-1} \right) + \frac{1}{2} \log |\mathbf{I} + \mathbf{K}_\beta \mathbf{B}| + \frac{1}{2} \mathbf{a}^T \mathbf{G}_\alpha \mathbf{a} - \mathbf{a}^T \mathbf{G}_{\alpha,\bar{\alpha}} \bar{\mathbf{a}}.$$

We then give the equations to compute the derivatives below. For compactness of notation, we use $\odot$ to denote element-wise product and use $\mathbf{1}$ to denote the vector of ones. In addition, we introduce a linear operator $\mathrm{diag}$ with overloaded definitions:

1. $\mathrm{diag} : \mathbb{R}^N \to \mathbb{R}^{N \times N}$ which constructs a diagonal matrix from a vector
2. $\mathrm{diag} : \mathbb{R}^{N \times N} \to \mathbb{R}^N$ which extracts the diagonal elements of a matrix to a vector.

**Proposition 2.** *The gradients of* $\mathrm{KL}[q||p]$ *is as follows:*

$$\nabla_\mathbf{a} \mathrm{KL}[q||p] = \mathbf{G}_\alpha \mathbf{a} - \mathbf{G}_{\alpha,\bar{\alpha}} \bar{\mathbf{a}}$$
$$\nabla_\alpha \mathrm{KL}[q||p] = \mathrm{diag}(\mathbf{a}) \left( \partial_\alpha \mathbf{G}_\alpha \mathbf{a} - \partial_\alpha \mathbf{G}_{\alpha,\bar{\alpha}} \bar{\mathbf{a}} \right)$$
$$\nabla_\mathbf{B} \mathrm{KL}[q||p] = \frac{1}{2} (\mathbf{I} + \mathbf{K}_\beta \mathbf{B})^{-1} (\mathbf{K}_\beta \mathbf{B} \mathbf{K}_\beta - \mathbf{\Delta}_\beta)(\mathbf{I} + \mathbf{B}\mathbf{K}_\beta)^{-1}$$
$$\nabla_\beta \mathrm{KL}[q||p] = \left( \partial_\beta \mathbf{K}_\beta \odot (\mathbf{B}^{-1} + \mathbf{K}_\beta)^{-1} \mathbf{G}_\beta (\mathbf{B}^{-1} + \mathbf{K}_\beta)^{-1} \right) \mathbf{1} - \left( \partial_\beta \mathbf{\Delta}_\beta \odot (\mathbf{B}^{-1} + \mathbf{K}_\beta)^{-1} \right) \mathbf{1}$$

*where* $\mathbf{\Delta}_\beta = \mathbf{G}_\beta - \mathbf{K}_\beta$ *and* $\partial$ *is defined as the partial derivative with respect to the left argument.*[15] *In particular, if the $p$ is normal,*

$$\nabla_\mathbf{a} \mathrm{KL}[q||p] = \mathbf{K}_\alpha \mathbf{a}$$
$$\nabla_\alpha \mathrm{KL}[q||p] = \mathrm{diag}(\mathbf{a}) \partial_\alpha \mathbf{K}_\alpha \mathbf{a}$$
$$\nabla_\mathbf{B} \mathrm{KL}[q||p] = \frac{1}{2} (\mathbf{I} + \mathbf{K}_\beta \mathbf{B})^{-1} \mathbf{K}_\beta \mathbf{B} \mathbf{K}_\beta (\mathbf{I} + \mathbf{B}\mathbf{K}_\beta)^{-1}$$
$$\nabla_\beta \mathrm{KL}[q||p] = \left( \partial_\beta \mathbf{K}_\beta \odot (\mathbf{B}^{-1} + \mathbf{K}_\beta)^{-1} \mathbf{K}_\beta (\mathbf{B}^{-1} + \mathbf{K}_\beta)^{-1} \right) \mathbf{1}$$

The derivation of Proposition 2 is simply mechanical, so we omit it here.

Here we only show the derivative with respect to $\mathbf{B}$. Suppose $\mathbf{B} = \mathbf{L}\mathbf{L}^T$. Then one can apply the chain rule and get

$$\nabla_{\mathbf{L}}\mathrm{KL}[q||p] = 2\nabla_B\mathrm{KL}[q||p]\mathbf{L}.$$

## B.2 Expected Log-Likelihood

### B.2.1 Evaluation

The evaluation of the expected log-likelihood depends on the mean and covariance in (8) , which we repeat here

$$\hat{m}_{|\mathbf{y}}^{\alpha}(x) = \mathbf{k}_{x,\alpha}\boldsymbol{a}, \qquad \hat{k}_{|\mathbf{y}}^{\beta}(x, x') = k_{x,x'} - \mathbf{k}_{x,\beta}\left(\mathbf{B}^{-1} + \mathbf{K}_{\beta}\right)^{-1}\mathbf{k}_{\beta,x'}.$$

Its derivation is trivial by the definition of $q$ in (16) and (18). For $N$ observations, the vector form $\hat{\mathbf{m}} \in \mathbb{R}^N$ and $\hat{\mathbf{s}} \in \mathbb{R}^N$ of the mean and the covariance above evaluated on each observation can be computed in $O(N)$ as

$$\hat{\mathbf{m}} = \mathbf{K}_{X,\alpha}\mathbf{a}$$
$$\hat{\mathbf{s}} = \mathrm{diag}\left(\mathbf{K}_X - \mathbf{K}_{X,\beta}(\mathbf{B}^{-1} + \mathbf{K}_{\beta})^{-1}\mathbf{K}_{\beta,X}\right)$$
$$= \mathrm{diag}(\mathbf{K}_X) - \left(\mathbf{K}_{X,\beta} \odot (\mathbf{K}_{X,\beta}(\mathbf{B}^{-1} + \mathbf{K}_{\beta})^{-1})\right)\mathbf{1}$$
$$= \mathrm{diag}(\mathbf{K}_X) - \left(\mathbf{K}_{X,\beta} \odot (\mathbf{K}_{X,\beta}\mathbf{B}(\mathbf{I} + \mathbf{K}_{\beta}\mathbf{B})^{-1})\right)\mathbf{1}.$$

Given $\hat{\mathbf{m}}$ and $\hat{\mathbf{s}}$, the expected log-likelihood can be evaluated either in closed-form for Gaussian likelihood or by sampling for general likelihoods.

### B.2.2 Gradients

The computation of the gradients of the expected log-likelihood can be completed in two steps. First, we compute the gradients of $\mathbb{E}_q[\log p_{\theta}(y|f)]$ with respect to $(\theta, \hat{\mathbf{m}}, \hat{\mathbf{s}})$ (i.e. $\nabla_{\hat{\mathbf{m}}}e$, $\nabla_{\hat{\mathbf{s}}}e$, and $\nabla_{\hat{\theta}}e$ ). Because $\log p_{\theta}(y|f)$ is the sum of $N$ terms, this step can be done in $O(N)$: for each observation $x$, let $q(f(x)) = \mathcal{N}(f(x)|\hat{m}, \hat{s})$ be a scalar Gaussian; under standard regularity conditions, we have

$$\nabla_{\hat{m}}\mathbb{E}_q[\log p_{\theta}(y|f(x))] = \mathbb{E}_q[\nabla_{\hat{m}}\log q(f(x))\log p_{\theta}(y|f(x))]$$
$$\nabla_{\hat{s}}\mathbb{E}_q[\log p_{\theta}(y|f(x))] = \mathbb{E}_q[\nabla_{\hat{s}}\log q(f(x))\log p_{\theta}(y|f(x))]$$
$$\nabla_{\theta}\mathbb{E}_q[\log p_{\theta}(y|f(x))] = \mathbb{E}_q[\nabla_{\theta}\log p_{\theta}(y|f(x))]$$

where $\nabla_{\theta}\log p_{\theta}(y|f(x))$ can be found, for example, in [22]. The above can be calculated in closed-form for Gaussian likelihood or by sampling for general likelihoods.

Next we propagate these gradients by chain rule. The results are summarized below.

**Proposition 3.** *Let $e = \mathbb{E}_q[\log p_{\theta}(y|f)]$. Suppose $k(x, x') = \rho^2 g_s(x, x')$ for some hyper-parameters $\rho, s \in \mathbb{R}$. The gradients of $e$ are as follows:*

$$\nabla_a e = \mathbf{K}_{X,\alpha}^T\nabla_{\hat{\mathbf{m}}}e$$
$$\nabla_{\alpha}e = \mathrm{diag}(\mathbf{a})\partial\mathbf{K}_{X,\alpha}^T\nabla_{\hat{\mathbf{m}}}e$$
$$\nabla_{\mathbf{B}}e = -(\mathbf{I} + \mathbf{K}_{\beta}\mathbf{B})^{-1}\mathbf{K}_{X,\beta}^T\mathrm{diag}(\nabla_{\hat{\mathbf{s}}}e)\mathbf{K}_{X,\beta}(\mathbf{I} + \mathbf{B}\mathbf{K}_{\beta})^{-1}$$
$$\nabla_{\beta}\hat{e} = 2(\partial\mathbf{K}_{\beta}^T \odot (\mathbf{\Omega}\mathrm{diag}(\nabla_{\hat{\mathbf{s}}}e)\mathbf{\Omega}^T))\mathbf{1} - 2(\mathbf{\Omega} \odot \partial\mathbf{K}_{\beta,X})\nabla_{\hat{\mathbf{s}}}e$$
$$\nabla_{\log\rho}e = \hat{\mathbf{m}}^T\nabla_{\hat{\mathbf{m}}}e + 2\hat{\mathbf{s}}^T\nabla_{\hat{\mathbf{s}}}e$$
$$\nabla_s e = (\partial_s\mathbf{K}_{X,\alpha}\mathbf{a})^T\nabla_{\hat{\mathbf{m}}}e - 2\mathbf{1}^T\left(\mathbf{\Omega} \odot \partial_s\mathbf{K}_{\beta,X}\right)\nabla_{\hat{\mathbf{s}}}e$$

*where $\mathbf{\Omega} = \mathbf{B}(\mathbf{I} + \mathbf{K}_{\beta}\mathbf{B})^{-1}\mathbf{K}_{\beta,X}$.*

The derivation of Proposition 3 is only technical, so we omit it here.

# C Experiment Setup

## C.1 The Covariance Function

For all the models, we assume the prior is zero mean and has covariance defined by a SE-ARD kernel [19]

$$k(x, x') = \rho^2 \phi_x^T \phi_x = \rho^2 \prod_{d=1}^{D} \exp\left(\frac{-(x_d - x'_d)^2}{2s_d^2}\right),$$

where $s_d > 0$ is the length scale of dimension $d$. For the variational posterior, we use the generalized SE-ARD kernel [2]

$$\psi_x^T \psi_{x'} = \prod_{d=1}^{D} \left(\frac{2l_{x,d}l_{x',d}}{l_{x,d}^2 + l_{x',d}^2}\right)^{1/2} \exp\left(-\frac{\|x_d - x'_d\|^2}{l_{x,d}^2 + l_{x',d}^2}\right), \tag{19}$$

where $l_{x,d} = s_d \cdot c_{x,d}$ is the length-scale parameter. That is, we evaluate

$$\mathbb{C}[L_m f(\tilde{x}_m), L_n f(\tilde{x}_n)] = \psi_{\tilde{x}_m}^T \psi_{\tilde{x}_n}$$

where the associated length-scalar parameters implicitly define the linear operators $L_m$ and $L_n$.

This kernel is first introduced in [26] by convoluting a SE-ARD kernel with Gaussian integral kernels, and later modified into its current form (19) in [2]. From (19), we see it contains SE-ARD as a special case. That is, $\psi_x = \phi_x$ when $c_{x,d} = 1, \forall d \in \{1, \ldots, D\}$. But in general $c_{x,d}$ can be a function of $x$. Therefore, it can be shown that $\psi_x$ spans an RKHS that contains the RKHSs spanned by $\phi_x$ for all length-scales, and every cross covariance can be computed as $\mathbb{C}[L_m f(\tilde{x}_m), f(x)] = \rho \psi_{\tilde{x}_m}^T \phi_x$.

Note: all the algorithms in our comparisons use this generalized SE-ARD kernel.

## C.2 Online Learning Procedure

Algorithm 1 summarizes the online learning procedure used by all stochastic algorithms (the algorithms differs only in whether the bases are shared and how the model is updated; see Table 1.), where each learner has to optimize all the parameters on-the-fly using *i.i.d.* data. The hyper-parameters are first initialized heuristically by median trick using the first mini-batch (in the GPR experiments, $s_d$ is initialized as the median of pairwise distances of the sampled observations; $\sigma^2$ is initialized as the variance of the sampled outputs; $\rho = 1$). We incrementally build up the variational posterior by including $N_\Delta \leq N_m$ observations in each mini-batch as the initialization of new variational basis functions (we initialize a new variational basis as $\tilde{x}_m = x_n$ and $c_{\tilde{x},d} = 1$, where $x_n$ is a sample from the current mini-batch). Then all the hyper-parameters and the variational parameters are updated online. These steps are repeated for $T$ iterations.

# D  Complete Experimental Results

## D.1  Experimental Results on KUKA datasets

|        | SVDGP    | SVI   | iVSGPR | VSGPR | GPR     |
|--------|----------|-------|--------|-------|---------|
| $Y_1$  | **0.985** | 0.336 | 0.411  | 0.085 | -3.840  |
| $Y_2$  | **1.359** | 0.458 | 0.799  | 0.468 | -23.218 |
| $Y_3$  | **0.951** | 0.312 | 0.543  | 0.158 | -8.145  |
| $Y_4$  | **1.453** | 0.528 | 0.906  | 0.722 | -0.965  |
| $Y_5$  | **1.350** | 0.311 | 0.377  | 0.425 | -0.990  |
| $Y_6$  | **1.278** | 0.367 | 0.631  | 0.559 | -0.639  |
| $Y_7$  | **1.458** | 0.425 | 0.877  | 0.886 | 0.449   |
| mean   | **1.262** | 0.391 | 0.649  | 0.472 | -5.335  |
| std    | **0.195** | 0.076 | 0.201  | 0.265 | 7.777   |

(a) Variational Lower Bound ($10^5$)

|        | SVDGP     | SVI   | iVSGPR | VSGPR | GPR   |
|--------|-----------|-------|--------|-------|-------|
| $Y_1$  | **0.058** | 0.186 | 0.165  | 0.171 | 0.257 |
| $Y_2$  | **0.028** | 0.146 | 0.095  | 0.126 | 0.249 |
| $Y_3$  | **0.058** | 0.195 | 0.133  | 0.181 | 0.298 |
| $Y_4$  | **0.027** | 0.124 | 0.088  | 0.114 | 0.198 |
| $Y_5$  | **0.028** | 0.195 | 0.178  | 0.132 | 0.243 |
| $Y_6$  | **0.034** | 0.178 | 0.137  | 0.140 | 0.224 |
| $Y_7$  | **0.028** | 0.155 | 0.099  | 0.108 | 0.146 |
| mean   | **0.037** | 0.169 | 0.128  | 0.139 | 0.231 |
| std    | **0.013** | 0.025 | 0.033  | 0.026 | 0.045 |

(b) Prediction Error (nMSE)

Table 3: Experimental results of KUKA$_1$ after 2,000 iteration. $Y_i$ denotes the $i$th output.

|        | SVDGP     | SVI   | iVSGPR | VSGPR | GPR     |
|--------|-----------|-------|--------|-------|---------|
| $Y_1$  | **1.047** | 0.398 | 0.631  | 0.399 | -3.709  |
| $Y_2$  | **1.387** | 0.450 | 0.767  | 0.515 | -31.315 |
| $Y_3$  | **0.976** | 0.321 | 0.568  | 0.232 | -12.230 |
| $Y_4$  | **1.404** | 0.507 | 0.630  | 0.654 | -1.026  |
| $Y_5$  | **1.332** | 0.317 | 0.378  | 0.511 | -0.340  |
| $Y_6$  | **1.260** | 0.368 | 0.585  | 0.538 | -0.221  |
| $Y_7$  | **1.405** | 0.437 | 0.519  | 0.918 | 0.526   |
| mean   | **1.259** | 0.400 | 0.583  | 0.538 | -6.902  |
| std    | **0.165** | 0.065 | 0.110  | 0.197 | 10.770  |

(a) Variational Lower Bound ($10^5$)

|        | SVDGP     | SVI   | iVSGPR | VSGPR | GPR   |
|--------|-----------|-------|--------|-------|-------|
| $Y_1$  | **0.056** | 0.168 | 0.126  | 0.151 | 0.281 |
| $Y_2$  | **0.026** | 0.147 | 0.102  | 0.124 | 0.248 |
| $Y_3$  | **0.056** | 0.194 | 0.127  | 0.179 | 0.325 |
| $Y_4$  | **0.029** | 0.127 | 0.127  | 0.110 | 0.186 |
| $Y_5$  | **0.029** | 0.189 | 0.170  | 0.125 | 0.232 |
| $Y_6$  | **0.035** | 0.181 | 0.144  | 0.144 | 0.232 |
| $Y_7$  | **0.034** | 0.152 | 0.166  | 0.104 | 0.133 |
| mean   | **0.038** | 0.166 | 0.137  | 0.134 | 0.234 |
| std    | **0.012** | 0.023 | 0.022  | 0.024 | 0.058 |

(b) Prediction Error (nMSE)

Table 4: Experimental results of KUKA$_2$ after 2,000 iterations. $Y_i$ denotes the $i$th output.

## D.2 Experimental Results on MuJoCo datasets

|       | SVDGP  | SVI   | iVSGPR | VSGPR | GPR         |
|-------|--------|-------|--------|-------|-------------|
| $Y_1$ | **7.373** | 3.195 | 5.948  | 4.312 | -22.256     |
| $Y_2$ | **6.019** | 2.141 | 3.905  | 2.328 | -45.351     |
| $Y_3$ | **6.350** | 2.543 | 4.695  | 2.991 | -147.881    |
| $Y_4$ | **5.852** | 2.417 | 4.792  | 2.468 | -23.999     |
| $Y_5$ | **6.280** | 2.609 | 5.316  | 3.622 | -8.626      |
| $Y_6$ | **5.152** | 1.043 | 4.418  | 3.452 | -11296.669  |
| $Y_7$ | **5.270** | 2.093 | 4.183  | 1.676 | -7745.055   |
| $Y_8$ | **6.471** | 2.585 | 5.040  | 3.068 | -47.540     |
| $Y_9$ | **5.293** | 0.979 | 2.592  | 1.482 | -73477.168  |
| mean  | **6.007** | 2.178 | 4.543  | 2.822 | -10312.727  |
| std   | **0.673** | 0.692 | 0.898  | 0.871 | 22679.778   |

(a) Variational Lower Bound ($10^5$)

|       | SVDGP  | SVI   | iVSGPR | VSGPR | GPR   |
|-------|--------|-------|--------|-------|-------|
| $Y_1$ | **0.049** | 0.087 | 0.067  | 0.088 | 0.133 |
| $Y_2$ | **0.068** | 0.163 | 0.112  | 0.122 | 0.196 |
| $Y_3$ | **0.064** | 0.134 | 0.091  | 0.112 | 0.213 |
| $Y_4$ | **0.073** | 0.144 | 0.087  | 0.121 | 0.179 |
| $Y_5$ | **0.068** | 0.132 | 0.080  | 0.103 | 0.159 |
| $Y_6$ | **0.094** | 0.251 | 0.107  | 0.131 | 0.253 |
| $Y_7$ | **0.084** | 0.168 | 0.100  | 0.145 | 0.348 |
| $Y_8$ | **0.063** | 0.132 | 0.087  | 0.104 | 0.178 |
| $Y_9$ | **0.088** | 0.255 | 0.165  | 0.131 | 0.258 |
| mean  | **0.072** | 0.163 | 0.099  | 0.118 | 0.213 |
| std   | **0.013** | 0.053 | 0.026  | 0.016 | 0.061 |

(b) Prediction Error (nMSE)

Table 5: Experimental results of MUJOCO$_1$ after 2,000 iterations. $Y_i$ denotes the $i$th output.

|       | SVDGP  | SVI   | iVSGPR | VSGPR | GPR          |
|-------|--------|-------|--------|-------|--------------|
| $Y_1$ | **7.249** | 3.013 | 6.429  | 4.161 | -33.219      |
| $Y_2$ | **5.994** | 2.475 | 4.800  | 2.770 | -23.276      |
| $Y_3$ | **6.239** | 2.258 | 4.819  | 3.044 | -59.757      |
| $Y_4$ | **5.935** | 2.093 | 4.489  | 2.547 | -27.259      |
| $Y_5$ | **6.387** | 2.452 | 5.457  | 3.725 | -1.786       |
| $Y_6$ | **7.320** | 1.087 | 4.639  | 4.043 | -24.198      |
| $Y_7$ | **5.346** | 1.754 | 3.947  | 1.667 | -255179.052  |
| $Y_8$ | **6.448** | 2.505 | 5.193  | 3.812 | -190.294     |
| $Y_9$ | **6.237** | 0.683 | 2.596  | 2.241 | -37673.328   |
| mean  | **6.350** | 2.036 | 4.708  | 3.112 | -32579.130   |
| std   | **0.586** | 0.699 | 0.993  | 0.825 | 79570.425    |

(a) Variational Lower Bound ($10^5$)

|       | SVDGP  | SVI   | iVSGPR | VSGPR | GPR   |
|-------|--------|-------|--------|-------|-------|
| $Y_1$ | **0.051** | 0.095 | 0.056  | 0.085 | 0.138 |
| $Y_2$ | **0.069** | 0.133 | 0.085  | 0.111 | 0.186 |
| $Y_3$ | **0.066** | 0.149 | 0.087  | 0.113 | 0.182 |
| $Y_4$ | **0.071** | 0.160 | 0.094  | 0.127 | 0.197 |
| $Y_5$ | **0.065** | 0.137 | 0.074  | 0.101 | 0.148 |
| $Y_6$ | **0.051** | 0.241 | 0.097  | 0.073 | 0.139 |
| $Y_7$ | **0.081** | 0.187 | 0.107  | 0.142 | 0.363 |
| $Y_8$ | **0.063** | 0.133 | 0.081  | 0.106 | 0.214 |
| $Y_9$ | **0.067** | 0.270 | 0.157  | 0.106 | 0.300 |
| mean  | **0.065** | 0.167 | 0.093  | 0.107 | 0.207 |
| std   | **0.009** | 0.053 | 0.026  | 0.019 | 0.072 |

(b) Prediction Error (nMSE)

Table 6: Experimental results of MUJOCO$_2$ after 2,000 iterations. $Y_i$ denotes the $i$th output.

## Footnotes

[14]The idea of combining the partially shared representation and the canonical parametrization is brought up by Hugh Salimbeni in our discussion at the conference.

[15]The additional factor of 2 is due to $\mathbf{K}_\beta$ is symmetric.