[Reviews · NeurIPS 2017]

Reviewer 1



The authors propose a method to improve the scalability of Gaussian process based models. Their method is based on using an approximation similar to the one typically used based on inducing points. The key difference is that the authors have a different number of inducing points in the expression for the approximate GP mean than in the expression for the approximate variance. Because the cost of computing the mean is much smaller than for the variance, the authors can then afford to use a much larger number of inducing points to approximate the mean, without introducing a significant additional cost. the result is much better predictive performance than traditional sparse methods based on inducing inputs. The experiments performed illustrate the gains of the proposed method. Quality The paper seems technically sound, although I did not examine the theoretical content in detail. Clarity The paper is clearly written. Originality The proposed method is original up to my knowledge. Significance The proposed method is highly significant. In my opinion it is one of the most relevant advances in the area of scalable Gaussian processes in the last years. Minor typos or comments: 1 - Figure 2, mujoco appears twice and kuka is missing. 2 - In line 67 it is said that the proposed approach over-estimates the variance. However it can be appreciated that it also underestimates it in other regions of the plot different from the right-most part.

Reviewer 2



In their paper the authors propose a sparse variational algorithm for Gaussian process inference which decouples the inducing points used for approximating mean and covariance of the posterior. As both time and space complexity scale linearly in the number of inducing points used for the mean, it can be increased to describe complex functions more accurately. Results on Gaussian process regression for robot data sets show a significant increase in the accuracy compared to other algorithms which use the same inducing points for modeling mean and covariance. The paper is well written. The authors explain the derivation of the SVDGP model in great detail and give a rather short, but sufficient description of the actual online learning algorithm. However, some details are not as clear as possible: I guess that GPR means Gaussian process regression on a subset of N = 1024 data points and VSGPR has only applied to a (larger) subset of the data points (section 4). And then GPR should not have any variational parameters, only the hyperparameters are updated by conjugate gradient (CG) (table 1). I recommend to clarify this and give the run times of the algorithms explicitly. Separating the inducing points for mean and covariance is a novel concept based on the RKHS representation of Gaussian processes. As this approach reduces the computational complexity, I expect that it will be useful for big data applications, where Gaussian process models are to slow without it. The changes proposed in the author feedback are suitable to improve the clarity of the paper.

Reviewer 3



Summary: The paper employs an alternative view of Titsias's sparse variational inference technique for Gaussian process regression, treating the mean function and covariance function as linear operators in RKHS. By decoupling the basis functions that each of these functions uses, the variational lower bound can now be computed in a O(NMa + NMb^2) complexity, linear in N, and linear in Ma (number of mean parameters) and quadratic in Mb (number of covariance parameters). Previous approaches have O(NM^2) complexity. This means one now can use much bigger Ma compared to Mb without significantly increasing the complexity. The catch is, however, the posterior prediction is no longer **calibrated** and as good, i.e. smaller Mb results in larger predictive variances. The method is compared against existing approaches on several regression datasets, showing lower predictive errors and achieving higher variational lower bounds. Details: The catch is that the predictive variance is large if Mb is small resulting in a poor predictive performance if measured using the log-likelihood. This can be seen in figure 1, and is mentioned in the text but has not been fully experimented (as the paper only uses the squared error metric which does not use the predictive variances). Unless I have missed it, the basis locations are not optimised but selected using several initial mini-batches. Would there be any problems if we optimise locations of the inducing points/bases as in Titsias's approach? The inducing points/inputs are grounded on some physical locations and can be plotted. Can we visualise the subspace parameterisation? For example, grounding some bases and plotting the samples generated from the corresponding GPs? And why is there a kink/sudden change in the traces for all methods in the middle of Figure 1? The learning curves for SVI are still going down after 2000 iterations -- does this use SNGD for the q(u)'s parameters? I think recent works on SVI for GP models (Hensman et al 2014, Bonilla et al 2014, 2015) all use SGD for the mean and Cholesky parameters. This seems to work well in practice, and avoid the need to deal with natural gradients or select a separate learning rate. It would be better to compare to these variants of SVI. Extensions to non-Gaussian likelihoods seem non-trivial as computing the expected log likelihood would decouple the variational parameters.